# REGen: Multimodal Retrieval-Embedded Generation for Long-to-Short Video Editing

**Weihan Xu[1]    Yimeng Ma[1]    Jingyue Huang[2]    Yang Li[1]    Wenye Ma[3]**
**Taylor Berg-Kirkpatrick[2]    Julian McAuley[2]    Paul Pu Liang[4]    Hao-Wen Dong[5]**

[1]Duke University    [2]University of California San Diego    [3]MBZUAI
[4]Massachusetts Institute of Technology    [5]University of Michigan
weihan.xu@duke.edu

## Abstract

Short videos are an effective tool for promoting contents and improving knowledge accessibility. While existing extractive video summarization methods struggle to produce a coherent narrative, existing abstractive methods cannot 'quote' from the input videos, i.e., inserting short video clips in their outputs. In this work, we explore new video editing models for generating shorts that feature a coherent narrative with embedded video insertions extracted from a long input video. We propose a novel retrieval-embedded generation (REG) framework that allows a large language model to quote multimodal resources while maintaining a coherent narrative. Our proposed *REGen* system first generates the output story script with quote placeholders using a finetuned large language model, and then uses a multimodal retrieval model to replace the quote placeholders by selecting a video clip that best supports the narrative from a pool of candidate quotable video clips. We examine the proposed method on the task of documentary teaser generation, where short interview insertions are commonly used to support the narrative of a documentary. Our objective evaluations show that the proposed method can effectively insert short video clips while maintaining a coherent narrative. In a subjective survey, we show that our proposed method outperforms existing abstractive and extractive approaches in terms of coherence, alignment, and realism in documentary teaser generation.

## 1   Introduction

Generating shorts from long videos allows audiences to digest information in a more engaging way and helps content creators promote their original contents. Unlike text or visual-only summaries, short videos with visuals and audio are more engaging [1], accelerate comprehension [2], and improve recommendation and search [3]. Existing approaches for producing shorts from long videos can be categorized into extractive or abstractive methods. Extractive methods stitch together video clips extracted from the input video [4–10], yet this may produce disjointed videos that do not together convey a coherent story. In contrast, abstractive approaches synthesize new narratives [11] and even new scenes [12], but these methods cannot insert extracted video clips from the input video to support the generated narrative. Moreover, while recent retrieval-augmented generation (RAG) methods can augment a large language model (LLM) with additional knowledge at inference time, these methods cannot quote multimodal materials from external sources and embed the exact quotes into their outputs, and they sometimes fabricate or misattribute content when faced with extended contexts [13–15].

Crafting effective short videos requires both creating a coherent narrative and grounding it with raw material extracts, especially for domains that necessitate strong factualness and reliability such

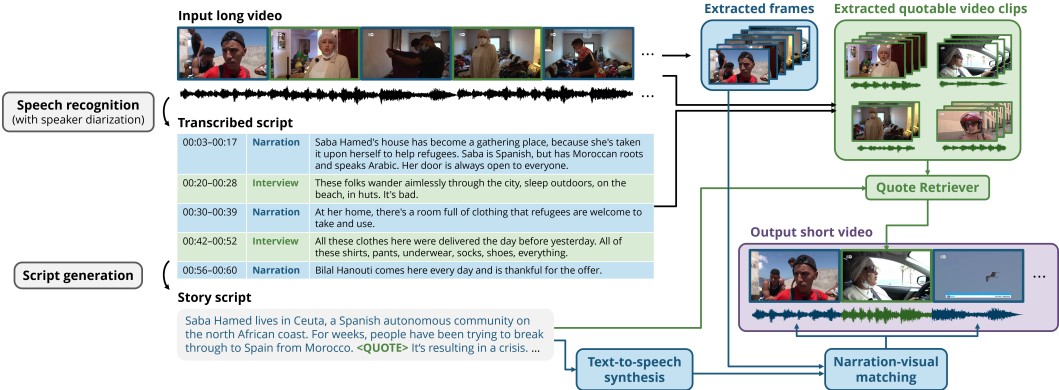

Figure 1: An overview of the proposed *REGen* system for long-to-short video editing. Given a long input video, we first transcribe the narrations and dialogues using a pretrained automatic speech recognition model, and then we use a finetuned large language model to generate the output story script with quote placeholders (i.e., the `<QUOTE>` token). For the generated narration, following [11], we first synthesize the narration into audio using a text-to-speech synthesis model and apply a narration-visual matching algorithm to find accompanying visuals. For the generated quote placeholders, we propose an encoder-decoder based *Quote Retriever* to select a video clip that best supports the narrative from a pool of quotable video clips extracted from the input video. The proposed system represents a new hybrid video editing model that combines abstractive and extractive methods.

as journalism and education. Further, there exists no video dataset with annotation that identifies externally quoted footage from original narrative segments, making it hard to approach this task through a data-driven approach.

In this work, we introduce *REGen*, a novel multimodal retrieval-embedded generation framework for editing long videos into shorts (see Fig. 1). In the first script generation stage, we finetune an LLM to generate story scripts with quote placeholders that will be fulfilled later in the second stage. In the second quotation retrieval stage, we then train a multitask encoder-decoder language model to select a video clip extracted from the input video so that it can support the narrative. For other generated narration, we follow [11] to synthesize the narration and accompanying it with visuals selected from frames extracted from the input video. To train the proposed system, we use the DocumentaryNet dataset [11] and construct training samples with transcribed, timestamped narrations and quotable interviews using an existing speech transcription and speaker diarization model [16].

We conduct extensive experiments to evaluate the effectiveness of our proposed models through objective evaluation metrics and a subjective survey. We show that the proposed REGen models can effectively edit a long documentary into a short teaser that has a coherent narrative and contain video quotations that support the narrative. Our experimental results show that our proposed method outperforms several abstractive and extractive baseline models in terms of coherence, audiovisual alignment, and realism. Video samples and all source code can be found on our website.[1]

Our contributions can be summarized as follows:

- We propose a new retrieval-embedded generation (REG) framework that allows an LLM to quote multimodal resources while maintaining a coherent narrative.
- We propose a novel long-to-short video editing model for generating shorts that feature a coherent narrative with embedded video insertions extracted from a long input video.

## 2   Related Work

**Generative modeling with factual grounding**   Previous work on generative modeling with factual grounding is primarily in the text domain and falls into two main categories: attribution-aware LLMs [17, 18] and retrieval-augmented generation (RAG) methods [19–21]. Attribution-aware LLMs enhance verifiability by generating responses with in-text citations or via post-hoc attributions. In

---

[1] https://wx83.github.io/REGen/

Table 1: Comparison of related methods in trailer generation and multimodal summarization

| Model | Type | | Input modality | | | Output modality | | | Video clip insertions[*] |
|---|---|---|---|---|---|---|---|---|---|
| | Ext. | Abs. | Frames | Video | Narr. | Text | Video | Frames | |
| CLIP-IT [24] | ✓ | ✗ | ✓ | ✓ | ✗ | ✗ | ✓ | ✓ | ✗ |
| A2Summ [4] | ✓ | ✗ | ✓ | ✓ | ✓ | ✓ | ✓ | ✓ | ✓ |
| LfVS [25] | ✓ | ✗ | ✓ | ✓ | ✓ | ✓ | ✓ | ✓ | ✓ |
| TGT [26] | ✓ | ✗ | ✓ | ✓ | ✗ | ✗ | ✓ | ✓ | ✗ |
| TaleSumm [27] | ✓ | ✗ | ✓ | ✓ | ✓ | ✓ | ✓ | ✓ | ✓ |
| VTSUM-BLIP [28] | ✗ | ✓ | ✓ | ✓ | ✗ | ✓[†] | ✓ | ✓ | ✗ |
| TeaserGen [11] | ✗ | ✓ | ✓ | ✓ | ✓ | ✓ | ✓ | ✓ | ✗ |
| REGen [11] | ✓ | ✓ | ✓ | ✓ | ✓ | ✓ | ✓ | ✓ | ✓ |

[*]Whether its outputs include extracted video clips with original sounds    [†]Achieved by dense video captioning

addition, they employ coarse attributions such as URLs [22] or document identifiers[23]. RAG-based approaches first retrieve relevant documents or text chunks and then condition the generation on the retrieved passages [19–21]. In contrast, our work targets multimodal outputs and performs exact quoting from external multimodal resources, generating narratives that directly quote raw quotable footage as grounding evidence to support the generated narratives.

**Long-to-short video editing**  Long-to-short video editing like video summarization or trailer generation addresses the challenge of condensing long-form videos into informative short videos. Prior work can be broadly divided into extractive and abstractive approaches. Extractive methods identify and splice together key clips directly from the source footage. For example, A2Summ [4] produces extractive summaries with a unified multimodal transformer-based model to predict key sentences and their time-aligned video segments. LfVS [25] utilizes large language models (LLMs) to extract key sentences from transcribed text, which are then paired with time-aligned video segments to create pseudo-ground-truth summaries. TaleSumm [27] introduces a two-level hierarchical model that identifies important sub-stories in a TV episode narrative. Although these techniques preserve the authenticity of the original clips, they often yield a disjoint viewing experience due to abrupt transitions between extracted segments. Abstractive methods, by contrast, first generate a cohesive narrative script and then retrieve or synthesize matching visuals. For instance, TeaserGen [11] prompts a large language model to produce a teaser script and subsequently fetch corresponding video clips. VTSUM-BLIP [28] jointly train parallel video and text summarization decoders, enabling end-to-end video-to-video summarization. Though abstractive approaches deliver smoother, more story-like short videos, they risk drifting from factual grounding. In this work, we propose a hybrid framework that automatically generates a coherent narrative and seamlessly inserts extractive quotable segments as grounding evidence. We compare related methods in Table 1.

## 3  Method

To generate short videos that quote contents from long videos, we adopt a two-stage method: first, we generate scripts with explicit quotation encoding; then we retrieve the corresponding quotable segments from long videos to fulfill each quotation coherently and support the surrounding narration.

### 3.1  Generating Script with Quotation via Fine-Tuned LLaMA

To train an LLM to identify quote insertion points, we leverage ASR with speaker diarization (see Appendix D) to generate data with quotable segments separated from narration. We explore two quote encoding methods for finetuning a pretrained language model to enable quoting from long contexts:

$$\text{REGen-DQ (direct quote)} : \qquad \ldots, x_i, \texttt{<SOQ>}, y_1, \ldots, y_n, \texttt{<EOQ>}, x_{i+1}, \ldots \qquad (1)$$

$$\text{REGen-IDQ (indirect quote)} : \qquad \ldots, x_i, \texttt{<QUOTE>}, x_{i+1}, \ldots \qquad (2)$$

To ensure comprehensive coverage of the source material, we transcribe the documentary audio with WhisperX [16], split it into ten chunks, and use GPT-4o [29] to generate one-sentence summary for

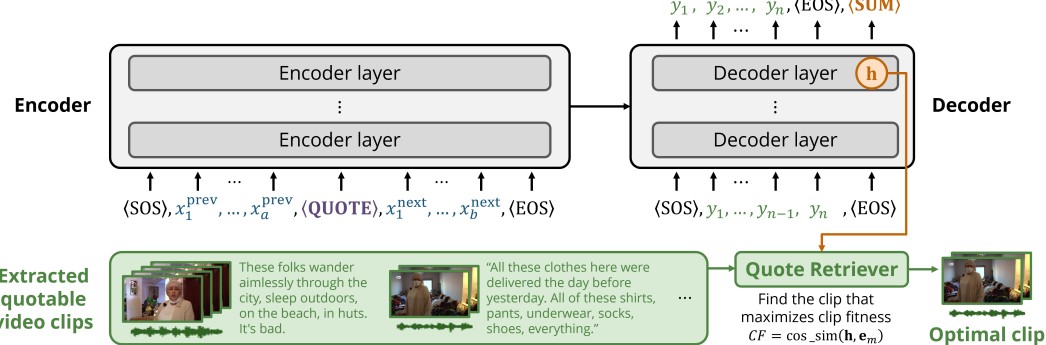

Figure 2: An illustration of the proposed two-stage quote retriever REGen-IDQ. We finetune an encoder-decoder language model that learns to 1) fulfill quotation placeholders (i.e., the <QUOTE> token) and 2) produce an embedding vector $\mathbf{h}$ that summarizes the quotation content. We then use the embedding vector $\mathbf{h}$ as the query to retrieve a video clip from a pool of candidate quotable video clips extracted from the input video. The optimal clip is selected based on our proposed *clip fitness* measure (see Section 3.2 for its definition). In this work, we consider all non-narrator video clips as candidates for the quote retriever. Note that this framework can be generalized to support quoting materials in any modality such as audio and images as long as we can find a proper fitness measure.

each chunk as inputs to LLM. Specially, we finetune LLaMA [30] using instruction finetuning. The finetuning templates can be found in Appendices B.4 and B.5.

For the generated narrations, following TeaserGen [11], we accompany the synthesized narration with visuals using an interval-based matching approach, extracting content corresponding to each narration sentence. We use the pretrained UniVTG model [31] to identify highlights aligned with the narrations. For the quotable part, we search our quotable clip base for the sentence embedding closest to the text wrapped in <SOQ> and <EOQ> for direct quotes. We will refer to this model as **REGen-DQ**. For indirect quotes, we propose a retriever module to retrieve quotable segments that blend with the surrounding narration (see Section 3.2), which we will refer to as **REGen-IDQ**.

## 3.2 Retrieving Quotable Segments Aligned with Narrative Context for RGEen-IDQ

As shown in Fig. 2, to retrieve a quotable segment from a curated database that supports the surrounding narrative, we frame this task as a multitask learning problem: the proposed quote retriever is trained jointly with a masked language modeling loss and a retrieval loss.

To fulfill the content of the quotation placeholders generated by REGen-IDQ, we finetune a pretrained encoder-decoder model (specifically, BART [32] in this work) to perform masked infilling conditioned on surrounding narrations. Let previous sentence be $s^{\mathrm{prev}} = \left(x_1^{\mathrm{prev}}, \ldots, x_a^{\mathrm{prev}}\right)$ and the following sentence be $s^{next} = (x_1^{\mathrm{next}}, \ldots, x_b^{\mathrm{next}})$. We have our encoder input and decoder output as

$$\text{Input}: \quad \text{<SOS>}, x_1^{\mathrm{prev}}, \ldots, x_a^{\mathrm{prev}}, \text{<QUOTE>}, x_1^{\mathrm{next}}, \ldots, x_b^{\mathrm{next}}, \text{<EOS>} \tag{3}$$

$$\text{Output}: \quad \text{<SOS>}, y_1, \ldots, y_n, \text{<EOS>}, \text{<SUM>} \tag{4}$$

The proposed method decodes meaningful sequence supporting nearby narrations, while the added special <SUM> token is expected to summarize the content of the quotation.

Inspired by retrieval-augmented generation (RAG) [19–21], we propose a retrieval module to find a suitable quotable segment from a candidate pool that best matches the decoded sentences. We use the hidden state of the final decoder token (i.e., the special <SUM> token) from each generated sequence to retrieve quotable video segments from the candidate pool. For each <QUOTE> placeholder and a pool of candidate quotable video clips $C = \{c_1, \ldots, c_M\}$, we retrieve the optimal video clip $c^*$ that maximizes the *clip fitness*, defined as $CF = \mathrm{cos\_sim}(\mathbf{h}, \mathbf{e}_m)$, where $h$ is the last-layer hidden state at the final decoder layer for the last token (i.e., the <SUM> token) and $\mathbf{e}_m$ is an embedding vector that captures the semantic meaning of a candidate clip $c_m$. We consider two variants of $\mathbf{e}_m$: First, we consider $\mathbf{e}_m = f(\mathrm{concat}(\mathbf{e}_m^{\mathrm{text}}, \mathbf{e}_m^{\mathrm{img}}))$, where $f$ is a learnable mapping parameterized as a two-layer multi-layer perceptrons and $\mathbf{e}_m^{\mathrm{text}}$ is the sentence embedding of the whole narration of the quote video segments. To aggregate visual information, we define $\mathbf{e}_m^{\mathrm{img}}$ as the concatenated

frame embeddings of three randomly selected frames for the quotable segments. The proposed multimodal fusion module $f$ is expected to learn to combine visual and textual information efficiently to optimize the retrieval performance. We will refer to this retriever as **QuoteRetriever-TV**. The REGen-IDQ model using this retriever will be referred to as **REGen-IDQ-TV**. In addition, we consider $\mathbf{e}_m = \mathbf{e}_m^{\text{text}}$, i.e., a retrieval model that considers only textual information. We will refer to this retriever as **QuoteRetriever-T** and the corresponding full system as **REGen-IDQ-T**.

During training, we jointly train the fusion module and fine-tune the pretrained BART[32] model with a multitask loss function: $L = L_{\text{gen}} + \alpha L_{\text{ret}}$, where $L_{\text{gen}}$ is the token level cross-entropy loss for masked language modeling, and $L_{\text{ret}}$ is the retrieval loss defined as

$$L_{\text{ret}} = -\sum_{k=1}^{K} \log \frac{\exp\big(\text{cos\_sim}(\mathbf{h}_k, \mathbf{e}^*)\big)}{\sum_{c_j \in C_k^-} \exp\big(\text{cos\_sim}(\mathbf{h}_k, \mathbf{e}_j)\big)} , \tag{5}$$

where $e^*$ is the sentence embedding of the ground truth narration and $C_k^-$ the set of negatives samples for quotation placeholder $k$. Furthermore, we train the retriever with in-batch negative sampling, and explore a GroupSampler module to construct hard negative samples, further separating ground-truth quotable video clips from the remaining quotable video clips in the video clip base (see Appendix E.4 for more details).

## 4 Experimental Setup

### 4.1 Dataset and Implementation Details

In this work, we use the DocumentaryNet [11] dataset in our experiments. DocumentaryNet contains 1,269 documentaries paired with their teasers from three reliable sources: DW Documentary, Public Broadcasting Service (PBS) and National Geographic. We perform speaker diarization to generate scripts using WhisperX [16] on DocumentaryNet. We detect narrator segments by assuming that narrations correspond to the longest transcribed audio segments (we examine the robustness of our speaker diarization and narrator identification in Appendix C). In this work, we consider all non-narrator video clips as quotable video clip candidates for the quote retriever. To validate our data annotations, we recruited four people to validate the ASR and narrator-identification results. We use an FPS (frame per second) of 1. We include details about the datasets and implementation in Appendices D and E.

### 4.2 Script Generation

In our first experiment, we evaluate the performance of our proposed method in generating coherent scripts with quotations, as well as its ability to conduct exact quotations from input videos.

**Baselines** We compare our model with script existing multimodal video summarization model [4], teaser generation model [11], and three LLM-based method.

- **Random Extraction**: We randomly sample sentences from main documentary transcription.
- **Extractive-then-Smoothing (ETS)**: We select two interviews whose content is closest to the video title and ask GPT-4o to connect the extracted interviews into a cohesive story. We include the prompts in Appendix B.7.
- **A2Summ** [4]: This baseline model uses an extractive method to select joint textual and visual segments with temporal correspondence. Since A2Summ can only process videos shorter than 300 seconds, we divide each video into ten chunks, select key segments from each chunk separately, and concatenate them together.
- **TeaserGen** [11]: This baseline model divides the audio transcription of long videos into ten chunks, generates a one-sentence summary for each, and instructs GPT-4o [29] to weave these summaries into a story-like teaser ending with a compelling question.
- **GPT-4o-DQ**: We transcribe each documentary with WhisperX [16], split the transcript into ten chunks, summarize each chunk, and concatenate the summaries into a single documentary-level summary. Using this summary as context, GPT-4o [29] generates an introduction with in-text quotations. See Appendix B.1 for the prompt.
- **GPT-4o-SP-DQ**: We split the speaker-labeled script processed in Section 4.1 into ten chunks and ask GPT-4o [29] to generate a concise, engaging summary for each chunk. We then concatenate

these ten summaries to reconstruct a complete, speaker-labeled script. We include our prompts in Appendix B.3.

- **LLaMA-DQ**: We prompt LLaMA[30] to generate an introduction script given a summary of the main documentary content and include in-text quotations. We include the prompt in Appendix B.2.

**Objective Evaluation Metrics** We evaluate our quotation insertion with three primary metrics. First, we measure *quotation density index* (**QDI**), i.e., the average number of quotes inserted per documentary. Second, we compute *quote coverage rate* (**QCR**), the proportion of test videos in which at least one quotation is correctly inserted. Third, we define *overlap ratio* (**OR**) to measure the overlap between direct quoted contents by large language model with the ground truth interviews as OR = #{overlap words}/#{words in matched interviews}.

## 4.3 Quote Retriever Evaluation

In this experiment, we evaluate whether the proposed retriever can retrieve the correct video clip in our test set and assess its generalizability when applied to LLM generated scripts.

**Baselines** To evaluate the effectiveness of our proposed quote retriever, we compare it against two baselines: random selection and GPT-based infilling of `<QUOTE>` with surrounding narrations.

- **Random Selection**: Randomly choose interview segments for insertion.
- **GPT-4o Infilling**: Given the preceding and succeeding narration chunks, we prompt GPT-4o [29] to generate content to fill the `<QUOTE>` position, and then retrieve the nearest neighbor in the sentence embedding space [33] from our interview base. A detailed analysis for this baseline can be found in Appendix I.

**Evaluation Metrics** For objective evaluation, we evaluate our retrieval stage using recall, reporting Recall@1, Recall@5 and Recall@10 on teasers in the test set. Recall@5 indicates that the correct segment appears among the top five retrieved interviews. To further assess the generalizability of our retriever when applied to LLM-generated scripts, we conduct a subjective evaluation. Twenty-one participants (11 evaluating version A and 10 evaluating version B) rate each inserted interview segment in the generated teasers on a five-point Likert scale, judging the effectiveness of the insertion based on how well it supports the surrounding claim and maintains natural flow. We include survey questions in Appendix K.

## 4.4 Documentary Teaser Generation

In this experiment, we measure our model performance in documentary teaser generation task.

**Baselines** We consider Random Extraction, Extractive-then-Smoothing, A2Summ [4], and Teaser-Gen [11] described in Section 4.2 as baselines. Additionally, for GPT-4o-DQ, we extract quoted segments within quotation marks and use nearest-neighbor retrieval to fetch matching visual clips from the interview pool; for GPT-4o-SP-DQ, we retrieve interview segments from the interview pool via nearest-neighbor search on script segments labeled as non-narrator content.

**Evaluation Metrics** We first measure the quality of the final script, where each `<QUOTE>` marker generated by the script-generation stage has been replaced by its retrieved interview segment in the retrieval stage. Specifically, we report ROUGE F1 [34], which measures n-gram overlap and sequence continuity between generated and reference teaser scripts. In addition, we assess narrative coherence using G-Eval [35] on the DeepEval platform [36]. In addition, following TeaserGen [11], we report five retrieval-based metrics: F1, scene change rate (SCR), repetitiveness (REP), CLIPScore [37], and VTGHLS [35, 11]. F1 measures retrieval accuracy against the ground-truth teaser, while SCR (the frequency of scene transitions) and REP (the degree of repetitive content) capture aspects of temporal continuity that affect the viewer experience. We report CLIPScore [37] (specifically, CLIPS-I and CLIP-N) to measure audiovisual alignment, and VTGHLS [35, 11] to measure the likelihood that each selected frame will be perceived as a highlight relevant to the video title (see Appendix F for more details). Moreover, we define the interview ratio as the fraction of interview time (in seconds) in a video.

Table 2: Objective evaluation results for documentary teaser script generation

| Model | Before fulfillment | | | After fulfillment | | | | |
|---|---|---|---|---|---|---|---|---|
| | Tokens | QCR (%) | QDI | Tokens | R-1 | R-2 | R-L | G-Eval |
| Random extraction | - | 98 | 11.71 | 235 | 0.27 | 0.04 | 0.12 | $0.56 \pm 0.02$ |
| ETS | - | 96 | 1.96 | 340 | 0.21 | 0.03 | 0.11 | $0.81 \pm 0.01$ |
| A2Summ [4] | - | 96 | 3.98 | 172 | 0.27 | 0.03 | 0.13 | $0.42 \pm 0.01$ |
| TeaserGen [11] | - | - | - | 304 | 0.21 | 0.03 | 0.11 | $0.85 \pm 0.01$ |
| GPT-4o-DQ | 292 | 98 | 4.02 | 402 | 0.22 | 0.05 | 0.12 | $0.77 \pm 0.01$ |
| GPT-4o-SP-DQ | 631 | 100 | 22.33 | 1372 | 0.13 | 0.03 | 0.07 | $0.75 \pm 0.01$ |
| REGen-DQ | 153 | **76** | **2.31** | 210 | **0.28** | **0.05** | **0.13** | $0.43 \pm 0.02$ |
| REGen-IDQ-T | 98 | 67 | 1.98 | 172 | 0.25 | 0.04 | 0.13 | $0.57 \pm 0.02$ |
| REGen-IDQ-TV | 98 | 67 | 1.98 | 179 | 0.25 | 0.04 | 0.13 | $\mathbf{0.59 \pm 0.01}$ |
| Ground truth | - | 82 | 3.02 | 121 | - | - | - | $0.62 \pm 0.03$ |

Table 3: Comparisons of quote retrieval methods

| Retriever | Similarity measure | Recall@1 (%) | Recall@5 (%) | Recall@10 (%) | Insertion effectiveness |
|---|---|---|---|---|---|
| Random | - | $0.00 \pm 0.00$ | $0.28 \pm 0.48$ | $7.22 \pm 5.54$ | $3.08 \pm 0.25$ |
| GPT-4o infilling | Text only | $2.78 \pm 0.48$ | $13.89 \pm 1.27$ | $22.50 \pm 1.44$ | $2.48 \pm 0.31$ |
| QuoteRetriever-T | Text only | **5.00** | **17.50** | **30.00** | $\mathbf{3.56 \pm 0.22}$ |
| QuoteRetriever-TV | Text+Visual | **5.00** | 15.00 | 23.33 | $3.49 \pm 0.26$ |

Following the subjective study in Section 4.3, we randomly select ten documentaries from the test set, divide them into two groups of five, and ask participants to evaluate the generated teasers on coherence, alignment, realism, and interview effectiveness using a five-point Likert scale.

## 4.5 Ablation Study

We include additional ablation study in Appendix L. We evaluate the effects of the the max length of BART tokenizer [32] (Appendix L.1), the alpha parameter for balancing losses (Appendix L.2), the use of GroupSampler during retriever training (Appendix L.3), the choice of loss function (Appendix L.4), and the position of the retrieval token <SUM> (Appendix L.5).

# 5 Results

## 5.1 Script Generation

In this experiment, we compare our model performance in generating scripts with quotations against the following baselines: GPT-4o-DQ, GPT-4o-SP-DQ, random extraction, A2Summ [4], and Teaser-Gen [11]. First, we evaluate whether LLMs such as LLaMA [30] and GPT-4o [29] can directly quote from long contexts, and we report our results in Appendix G. We find that GPT-4o cannot quote exact content from long inputs when fed with the full transcript. In addition, while vanilla LLaMA cannot produce meaningful quotes using quotation marks, the proposed finetuning method with <SOQ> and <EOQ> increases the overlap ratio from 0.0 to 0.07. Second, we compare our model to GPT-4o-DQ, GPT-4o-SP-DQ, Random Extraction, A2Summ [4], and TeaserGen [11] using the quotation density index (QDI) and quote coverage rate (QCR). As shown in Table 2, we find that REGen-DQ achieves QCR and QDI values closest to those of the ground truth, indicating that REGen-DQ generates scripts with a similar quote distribution to the ground truth scripts.

## 5.2 Quote Retriever

In this experiment, we aim to compare the retrieval capability of our model against GPT-based infilling method. First, we report in Table 3 recalls for retrieving the correct interview segments given

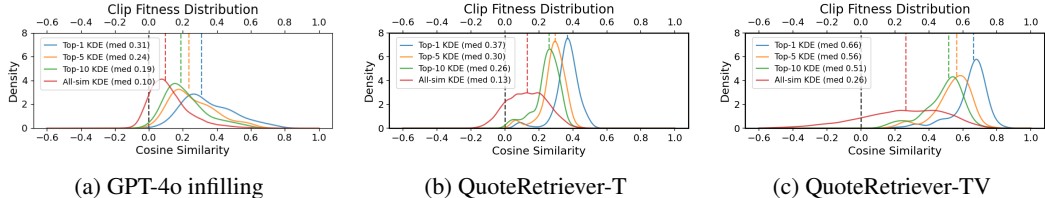

| | | (a) GPT-4o infilling | (b) QuoteRetriever-T | (c) QuoteRetriever-TV |

Figure 3: Comparison of infilling methods. The dotted lines indicate the median values.

Table 4: Objective evaluation results for documentary teaser generation

| Model | Dur (sec) | Interview ratio (%) | F1 (%) | SCR (%) | REP (%) | VTGHLS | CLIPS-I | CLIPS-N |
|---|---|---|---|---|---|---|---|---|
| Random extraction | 101 | 56 ± 20 | 1.10 | 20.71 | 0.41 | 0.83 | 0.55 | 0.62 |
| ETS | 142 | 34 ± 16 | 1.92 | 13.65 | 4.49 | 1.06 | 0.64 | 0.60 |
| A2Summ [4] | 73 | 42 ± 25 | 1.70 | 14.20 | 1.73 | 0.89 | 0.56 | 0.63 |
| TeaserGen [11] | 155 | - | 1.64 | **22.61** | 21.38 | 0.80 | - | 0.67 |
| GPT-4o-DQ | 151 | 42 ± 42 | 1.56 | 16.55 | 20.75 | 1.01 | 0.58 | 0.42 |
| GPT-4o-SP-DQ | 619 | 61 ± 17 | **2.07** | 12.38 | 18.33 | 1.02 | 0.62 | 0.62 |
| REGen-DQ | 95 | 37 ± 26 | 1.45 | 19.13 | 10.35 | 1.05 | 0.48 | 0.57 |
| REGen-IDQ-T | 77 | 35 ± 31 | 1.89 | 19.79 | 10.02 | 1.03 | **0.41** | **0.57** |
| REGen-IDQ-TV | 81 | 35 ± 31 | 1.90 | 19.86 | **9.70** | 1.02 | 0.39 | 0.57 |
| Ground truth | 76 | 54 ± 37 | 69.00* | 27.60 | >7.86 | <0.98 | 0.43 | 0.57 |

*Following [11], for each frame in the teaser, we retrieve the top 20 most similar frames from the main content using CLIP embeddings. We then apply pixel-by-pixel comparison to the 20 candidates; however, this strict matching may fail to identify identical frames due to the low frame rate.

their preceding and succeeding narration of each teaser in our test set. On average, each video in our test set has 66 candidate interviews with a standard deviation of 37. Our proposed QuoteRetriever-T and QuoteRetriever-TV outperform infilling with GPT-4o and random selection in terms of retrieval recall. See Appendix F for details about the objective evaluation on quote retrievers.

Second, to evaluate the discriminative capability of our retriever on interview segments, we plot in Fig. 3 the top-1 retrieval similarity ("Top-1 KDE") alongside the all similarity distribution ("All-sim KDE"), where 'all' refers to the similarity between the embedding output by the fine-tuned BART [32] model and all interview segments. We observe a more prominent separation between the top-1 KDE and All-sim KDE for QuoteRetriever-TV than the GPT-4o infilling approach, which is beneficial for more accurate retrieval performance due to the increased contrast between probable candidates.

Third, we evaluate quote retriever performance by measuring how effectively retrieved segments can be inserted into generated narration scripts in a subjective study. We report the mean score and 95% confidence interval in Table 3. Both QuoteRetriever-T and QuoteRetriever-TV outperform the GPT-infilling baseline in terms of interview effectiveness, indicating that pretrained GPT-4o cannot produce meaningful text to fulfill the <QUOTE> placeholders and support accurate retrieval.

### 5.3 Documentary Teaser Generation

To evaluate the performance of the proposed method for the documentary teaser generation task, we compare our proposed method against several baselines models: random extraction, A2Summ [4], TeaserGen [11], Extractive-then-Smoothing (ETS), GPT-4o-DQ, and GPT-4o-SP-DQ. As reported in Table 2, we can see that REGen-DQ achieves the highest ROUGE scores, indicating that REGen-DQ generates scripts closest to the ground-truth teasers Appendix J. We also find that REGen-IDQ-TV achieves the closest G-Eval [35] score to that of the ground-truth, and that GPT-4o-SP-DQ achieves the highest F1 score. Notably, the CLIPScore for the interview scenes of the ground truth is lower than that of the narration scenes. This suggests a lower narration-visual correspondence for interview scenes, which is partly because interview scenes usually focus on the interviewees rather than having visuals to support the narration content. Our proposed REGen models also result in a smaller CLIPS-I value than CLIPS-N, which is consistent to the ground truth documentary teasers. As can be seen from Table 5, our subjective evaluation results show that the proposed REGen-IDQ-TV model achieves the

Table 5: Subjective evaluation results for documentary teaser generation

| Model | Coherence↑ | Alignment↑ | Realness↑ | Interview effectiveness↑ |
|---|---|---|---|---|
| A2Summ [4] | $2.72 \pm 0.24$ | $2.87 \pm 0.26$ | $2.67 \pm 0.23$ | $3.07 \pm 0.24$ |
| TeaserGen [11] | $3.22 \pm 0.23$ | $2.92 \pm 0.24$ | $2.86 \pm 0.23$ | - |
| GPT-4o-SP-DQ | $3.08 \pm 0.24$ | $3.23 \pm 0.25$ | $2.81 \pm 0.25$ | $3.32 \pm 0.25$ |
| REGen-DQ | $2.97 \pm 0.27$ | $3.03 \pm 0.27$ | $2.75 \pm 0.30$ | $\textbf{3.33} \pm \textbf{0.29}$ |
| REGen-IDQ-TV | $\textbf{3.29} \pm \textbf{0.24}$ | $\textbf{3.30} \pm \textbf{0.26}$ | $\textbf{3.05} \pm \textbf{0.25}$ | $3.25 \pm 0.30$ |

highest scores in terms of coherence, alignment, and realness, outperforming our another proposed REGen-DQ model. Meanwhile, our proposed REGen-DQ method achieves the highest interview effectiveness score, but this does not reach significance difference against REGen-IDQ-TV.

For the extractive-then-smoothing (ETS) baseline, its higher VTGHLS score likely results from selecting interview chunks closest to the video title in sentence-embedding space [33], but this value is higher than the ground-truth VTGHLS. Additionally, the ETS method has a much lower scene change rate than the ground truth teaser. In Table 2, we observe that this method yields lower ROUGE scores than our proposed model, which indicates less overlap with the ground-truth teasers, while its higher G-Eval rating partly results from the enforced story-like prompt in Appendix B.7. Second, A2Summ [4] yields the lowest G-Eval score in Table 2, reflecting its low narrative cohesion, which is further verified by the lower coherence score in the subjective evaluation in Table 5. As shown in Table 4, our model produces scene change rate (SCR) and repetition rate (REP) closer to those of the ground truth than A2Summ, which aligns with the higher perceived realness of our model in the subjective test. Third, TeaserGen [11] achieves the highest G-Eval score of 0.85 in Table 2, indicating that it produces the most story-like and cohesive output; however, this value is higher than the ground truth value (0.62). Also, REGen-IDQ-TV achieves higher alignment than TeaserGen in the subjective test, suggesting that TeaserGen may select content that is less audio–visually aligned than naturally extracted interview segments. In addition, TeaserGen has the lowest VTGHLS score, suggesting that its retrieved frames are less likely to be considered as highlighted moments for its video title.

Finally, we compare our proposed model with GPT-based models, including GPT-4o-DQ and GPT-4o-SP-DQ. We can see that the proposed REGen models produce scripts with higher ROUGE scores, indicating that the generated scripts are closer to the ground-truth teaser scripts. In the documentary teaser generation task, although GPT-4o-SP-DQ achieves the highest F1 score, it exhibits a much lower scene change rate and a significantly longer teaser length than the ground truth. Even though we cap teaser length at 500 tokens (approximately 2.5 minutes of speech assuming a normal speaking pace of 150 wpm), the generated teasers remain substantially longer than real ones. The lower scene change rate and higher repetitiveness suggest that GPT-4o-SP-DQ selects repetitive video clips, which can lead to a negative viewer experience. This aligns with the lower perceived realness of GPT-4o-SP-DQ compared to REGen-IDQ-TV in Table 5. We also compare our model with GPT-4o-SP-TV, where we use the speaker-labeled script described in Section 4.1 and, with QuoteRetriever-TV, retrieve only when the speaker is labeled non-narrator. We report results in Appendix H.

## 6 Limitations and Future Work

We want to point out several notable limitations of our work. First, while including video insertions may improve the factualness grounding of the output videos, the proposed method still has the risk of misplacing a quote in a wrong context. This may be alleviated by grounding the first-stage script generation model with information about all the quotable materials so that it can better generate a more cohesive narrative. To examine this hypothesis, we include in Table 7 a baseline model (GPT-4o-DQ-NS) that supplies GPT-4o [29] with the full transcript and all candidate quotable interview segments for script generation. However, this is technically challenging for LLaMA-based models due to the their limited context-window [30], which prevents us from providing all the quote candidates as the context for script generation. Moreover, the proposed method relies on successful segmentation of the input video, which is possible in our case through speaker diarization that might not be applicable to other domains such as lecture recordings. For future work, we note that the proposed framework can be generalized to support quoting materials in other modalities as long as we can find a proper fitness measure. We plan to investigate quoting audio and images towards a

more capable video editing model. We would also like to extend our proposed retrieval embedded augmentation (REG) framework to other application domains such as virtual assistants, chatbots, and recommender systems.

## 7  Broader Impacts

We envision our proposed method to be applied to other fields such as education technology, natural language processing and information retrieval. For education technology, we believe our proposed model can be adapted to generate short review videos from lecture recordings to enhance learning experience. The proposed multimodal retrieval-embedded generation method can also be applied to current LLMs so that they can quote external sources embedded in their outputs. From the information retrieval perspective, our proposed method brings the power of LLMs to extractive methods where we can generate a coherent narrative to connect multiple extracted materials. We believe our proposed framework will contribute to improving knowledge accessibility by generating engaging short videos for long videos that may be more approachable to certain groups.

## 8  Conclusion

We have proposed a novel retrieval-embedded generation (REG) framework that allows an LLM to include multimodal quotations in its outputs. We have examined the proposed method on the task of documentary teaser generation. We have shown through objective and subjective evaluations the effectiveness of our proposed REGen models on quoting short interview clips within a coherent narrative. The subjective evaluations show that our proposed hybrid approach outperforms several abstractive and extractive baseline models in terms of coherence, alignment, and realism. Our work contributes towards next generation AI-assisted video editing tools that can assist video creators in selecting, cutting, and arranging raw video materials into a cohesive video.

## 9  Acknowledgment

This work is supported by the NVIDIA Academic Grant Program under the project titled "Teaser Generation for Long Documentaries and Educational Videos."

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

# A Comprehensive View of REGen System

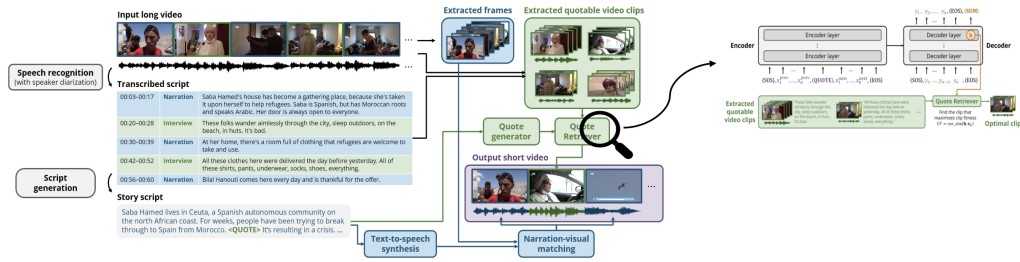

Figure 4: Comprehensive view of our REGen System

# B GPT-4o Prompts

## B.1 GPT-4o-DQ

You are a helpful assistant. Generate an engaging introduction of the content with quotation based on the following input

```
Input: f"{Ten sentences chunk summary}"
```

Output:

## B.2 LLaMA-DQ

You are a helpful assistant. Generate an engaging introduction of the content with quotation based on the following input

```
Input: f"{Ten sentences chunk summary}"
```

Output:

## B.3 GPT-4o-SP-DQ

A. You are a helpful assistant. Generate an engaging introduction of the content with quotation based on the following input.

B. "Generate a concise version of the following screenplay segment by preserving its plain text format " "where each line starts with either 'Narration:' or 'SpeakerID: [Speaker]:'. " "Limit the summary to approximately 50 tokens. "

## B.4 LLaMA Finetuning Template for REGen-DQ

Instruction: "You are a helpful assistant. Generate an engaging introduction of the content with quotation based on the following input."

```
Input: f"{Ten sentences chunk summary}"
```

Output:

$$\ldots, x_k, \texttt{<SOQ>}, y_1, \ldots, y_n, \texttt{<EOQ>}, x_{k+1}, \ldots \tag{6}$$

### B.5 LLaMA Finetuning Template for REGen-IDQ

Instruction: "You are a helpful assistant. Generate an engaging introduction of the content with quotation based on the following input."

Input: f"{Ten sentences chunk summary}"

Output:

$$\ldots, x_k, \texttt{<QUOTE>}, x_{k+1}, \ldots \qquad (7)$$

### B.6 G-Eval

"Assess how naturally the text flows in a story-like manner, evaluating grammatical correctness, syntactic variety, and seamless transitions that enhance narrative coherence."

### B.7 Extraction-then-Smoothing

System: You are a creative storyteller and writing coach.

User: Process the following input. It contains two interview chunks, each prefixed with "Interview:". Treat each "Interview: ..." segment as a single, indivisible unit and do not split or merge them.

Input: f"{Selected Interviews}"

Your task: 1. Generate a concise, engaging story that preserves the exact wording of each interview chunk. 2. Clearly indicate where in the narrative each interview chunk is inserted by using [Interview]. Now, please craft the story with these guidelines. """

## C  Robustness of Speaker Diarization

Our system demonstrates robust speaker diarization performance on our documentary dataset, achieving a narrator-prediction F1 score of 71.6%, start-time correctness of 93.5%, end-time correctness of 94.9%, and transcription accuracy of 94.9%. We thank the reviewer for pointing out the dependency of our framework on diarization quality. Although we acknowledge that alternative methods may further improve segment extraction on other datasets, we evaluated our framework's robustness by injecting controlled noise: we introduced varying proportions of segments labeled as "narrator" into the set of quotable interview segments. Using a fixed random seed, we then measured recall under different noise levels and reported in the table below; as expected, recall declines slightly as more incorrect segments are added.

Table 6: Robustness of Speaker Diarization on Downstream Task

| Model | Level of Accuracy | Recall@1 (%) | Recall@5 (%) | Recall@10 (%) |
|---|---|---|---|---|
| Ours | Build upon pretrained model | 3.33 | 17.50 | 31.67 |
| 10% extra error | Build upon pretrained model, intentionally add 10% narration | 1.67 | 14.17 | 29.17 |
| 20% extra error | Build upon pretrained model, intentionally add 20% narration | 0.83 | 12.50 | 27.50 |

## D  Dataset

**Generating Script with Speaker Annotation**   In order to generate scripts with timestamp and speaker labeled, following [38], we use WhisperX [16] for speaker diarization, which generates a script with start, end timestamps, speaker IDs as well as the transcribed text. We assume that narration typically dominates and corresponds to the longest audio track; therefore, we label the speaker with the longest transcript as the narrator. An example of our processed data is available on our demo page.[1] We constructed 941 paired teaser–main documentary screenplays from DocumentaryNet [11], and we include an additional 157 teaser-only samples. We estimate that in our training and validation sets, 766 out of 1,098 teaser part contain inserted videos, with an average of 1.8 inserts per example.

**Validating Automatic Speech Recognition and Narrator Identification Results**  We recruited four people to evaluate the annotation quality of our dataset. We asked them to assess if the automatically detected narrator was correct, if the start and end times of an interview segment were accurate, and if the transcription achieved over 95% accuracy. We provide an example of our annotation process on our demo page.[1] In the teaser part of our test set, we report the narrator prediction F1 score, audio-track start-time correctness, end-time correctness, and transcription accuracy. The narrator prediction F1 score is 71.6%. Start-time correctness is 93.5%. End-time correctness is 94.9%. Transcription accuracy is 94.9%. We present 128 interview segments with correct start times, end times, and transcriptions. In addition, 40 of 49 teasers in our test set include a second speaker, and on average each test video has 3.02 inserted clips. In our main documentary part of our test set, we also report the narrator prediction F1 score, audio-track start-time correctness, end-time correctness, and transcription accuracy. The narrator prediction F1 score is 88.7%. Start-time correctness is 90.8%. End-time correctness is 91.7%. Transcription accuracy is 96.3%. We present 2,472 interview segments with human-validated start times, end times, and transcriptions.

# E  Implementation Details

All experiments are conducted on a single NVIDIA A100 GPU with a batch size of 64. We reserve 5 % of the dataset (49 documentaries) for final testing and allocate 10 % of the remaining samples for validation. The learning rate is set to $1 \times 10^{-5}$, and training terminates when either the generation or the retrieval validation loss does not decrease within 30 consecutive epochs. We use Adam optimizer for training.

## E.1  Script Generation

In script generation stage, we remove any teaser outputs containing non-English tokens. Consequently, we fine-tune LLaMA [30] on 839 paired examples.

## E.2  Quote Retriever

In the quote retriever stage of our pipeline, we construct 47,883 training samples to jointly fine-tune the all-mpnet-base-v2 sentence embedder [33] and the Facebook BART-base model [32] with a maximum input length of 256 tokens. This extended context window enables the retriever to capture long-range dependencies—such as narrative shifts and cross-segment entity references in long main documentaries—that often exceed 128 tokens. For the documentary teaser-generation task (Section 4.4) and the accompanying subjective evaluation (Section 4.3), we cap BART's input window at 128 tokens. This choice is informed by our analysis in Table 2, which shows that the typical teaser length is 121 tokens; thus, a 128-token limit tightly encompasses almost all real-world examples. Moreover, reducing the context window cuts inference time and memory usage by nearly half during large-scale A/B studies, while ensuring that all generated outputs are compared under identical input constraints. In our GroupSampler module, if the number of distinct documentaries in a batch is less than the batch size, additional negatives are sampled from other documentaries to encourage fine-grained discrimination. When multiple interview segments occur between two consecutive narration chunks, we select the nearest preceding and succeeding narration scripts to construct each training sample.

## E.3  Documentary Teaser Generation

When constructing teasers with our proposed method, to prevent repetition, we maintain a sliding window over the last three selected clips and disallow any duplicate segment within that window.

## E.4  GroupSampler

If the number of distinct documentaries in a batch is less than the batch size, additional negatives are sampled from other documentaries to encourage fine-grained discrimination.

# F  Objective Evaluation Details

In Table 4, for each narration we may select multiple intervals (frames) to accompany it. We compute the CLIPScore[37] between the narration script and each frame, then use the highest CLIPScore among them as the score for that narration. For each interview segment, we similarly consider the highest CLIPScore within its interval. We report CLIPS-I as the CLIPScore measuring audiovisual alignment of interview segments and CLIPS-N as the CLIPScore measuring audiovisual alignment of narration segments.

We observe that certain interview segments in the main documentary are post-processed (e.g., trimmed or reshoot) before being incorporated into teasers. We treat these edited segments as distinct items in our retriever stage. To ensure a fair comparison with fully extractive baselines, we remove those interviews that cannot be exactly found in main documentary when compute F1, Repetitivenss, CLIPScore and VTGHLS when evaluating teaser generation task in Table 4. There is no difference in SCR because one interview is usually considered as one scene.

In Table 3, we also notice that some of the teaser are fully narrations and those are removed for retriever stage evaluation. Moreover, we conduct the experiments with 3 random seeds and report the standard deviation for random selection and GPT-4o infilling.

In Table 2, we run G-Eval with three random seeds and report the mean score and standard deviation.

# G  Script Generation Evaluation on Direct Quote from Long Contents

Table 7: Comparison in the capability of direct quotation

| Model | Quotation encoding | Summarized input script | Semantic similarity with the nearest neighbor | Overlap ratio |
|---|---|---|---|---|
| GPT-4o-DQ | Quotation marks | ✓ | 0.52 | 0.07 |
| GPT-4o-DQ-NS | Quotation marks | ✗ | 0.50 | 0.13 |
| GPT-4o-SP-DQ | Screenplay-like | ✓ | 0.69 | 0.17 |
| LLaMA-DQ | Quotation marks | ✓ | - | 0.00 |
| REGen-DQ | `<SOQ>` & `<EOQ>` | ✓ | 0.45 | 0.07 |

In Table 7, we report the semantic similarity between segments predicted as quotations—that is, LLM outputs enclosed by quotation markers—and their nearest neighbors in our interview database, as well as the Overlap Ratio defined in Section 4.2. When we provide GPT-4o [29] with the full documentary transcript, the overlap ratio is only 0.13. To accommodate extremely long inputs, we feed GPT-4o a summarized version of the main documentary; this further lowers the overlap to 0.07. The screenplay-like quotation encoding with GPT-4o raises the token overlap ratio to 0.17, but this remains inadequate. While vanilla LLaMA [30] cannot produce meaningful quotation markers, fine-tuning it with `<SOQ>` and `<EOQ>` increases the overlap ratio from 0 to 0.07. However, we note that this overlap ratio is still lower than that of the GPT-4o-based model. We expect better performance if we scale up the dataset.

## H    Additional Teaser Generation Evaluation

Table 8: Objective Evaluation for Teaser Generation Task

| Model | Dur (sec) | Interview Ratio (%) | F1 (%) | SCR (%) | REP (%) | VTGHLS | CLIPS-I | CLIPS-N |
|---|---|---|---|---|---|---|---|---|
| Random extraction | 101 | $56 \pm 20$ | 1.10 | 20.71 | 0.41 | 0.83 | 0.55 | 0.62 |
| ETS | 142 | $34 \pm 16$ | 1.92 | 13.65 | 4.49 | 1.06 | 0.64 | 0.60 |
| A2Summ [4] | 73 | $42 \pm 25$ | 1.70 | 14.20 | 1.73 | 0.89 | 0.56 | 0.63 |
| TeaserGen [11] | 155 | - | 1.64 | **22.61** | 21.38 | 0.80 | - | 0.67 |
| GPT-4o-DQ | 151 | $42 \pm 42$ | 1.56 | 16.55 | 20.75 | 1.01 | 0.58 | 0.42 |
| GPT-4o-SP-DQ | 619 | $61 \pm 17$ | **2.07** | 12.38 | 18.33 | 1.02 | 0.62 | 0.62 |
| GPT-4o-SP-TV | 673 | $64 \pm 17$ | 1.61 | 11.29 | 41.46 | 1.02 | 0.64 | 0.62 |
| REGen-IDQ (random) | 82 | $32 \pm 33$ | 1.34 | 20.40 | **7.50** | 1.03 | 0.41 | 0.57 |
| REGen-DQ | 95 | $37 \pm 26$ | 1.45 | 19.13 | 10.35 | 1.05 | 0.48 | 0.57 |
| REGen-IDQ-T | 77 | $35 \pm 31$ | 1.89 | 19.79 | 10.02 | 1.03 | 0.41 | 0.57 |
| REGen-IDQ-TV | 81 | $35 \pm 31$ | 1.90 | 19.86 | 9.70 | 1.02 | 0.39 | 0.57 |
| Ground truth | 76 | $54 \pm 37$ | 69.00* | 27.60 | 7.86 | <0.98 | 0.43 | 0.57 |

* Following [11], for each frame in the teaser, we retrieve the top 20 most similar frames from the main content using CLIP embeddings. We then apply pixel-by-pixel comparison to the 20 candidates; however, this strict matching may fail to identify identical frames due to the low frame rate.

When comparing GPT-4o-SP-DQ with GPT-4o-SP-TV, and GPT-4o-SP-TV with REGen-IDQ-TV, we observe a significant drop in teaser-generation performance for GPT-4o-SP-TV in Table 8, indicating that the surrounding narration produced by our fine-tuned script aids the our retrieval stage.

## I    GPT-4o Infilling Analysis

**Ground Truth**
The Mediterranean was once a major crossroads at the heart of the ancient world. Today, it has become a barrier separating Europe from Africa. Is there anything left of a past once shared? And what do today's distinct cultures have in common? Journalist Zineb Elmasrahr and Jafar Abdelkarim travel the coasts of the Mediterranean in search of answers. [Do you see yourself as a Tunisian Jew? Yes, with all the rights and responsibilities.] How can you afford the food for all these animals? God helps us. Join us to get to know the people and their dreams. A Mediterranean journey.

**GPT-4o Input**
The Mediterranean was once a major crossroads at the heart of the ancient world. Today, it has become a barrier separating Europe from Africa. Is there anything left of a past once shared? And what do today's distinct cultures have in common? Journalist Zineb Elmasrahr and Jafar Abdelkarim travel the coasts of the Mediterranean in search of answers. <mask>. How can you afford the food for all these animals? God helps us. Join us to get to know the people and their dreams. A Mediterranean journey.

*Here <mask> is used to represent quote placeholder as the input to LLM.*

**GPT-4o Output First Run**
The text that should replace `<mask>` is not explicitly provided in the query. However, based on the context, it seems like the `<mask>` placeholders are meant to be filled with phrases or sentences that connect the narrative. Here is a suggestion for what could replace the `<mask>` placeholders:
1. "How can you afford the food for all these animals? God helps us." - This seems to be a direct quote from someone they meet on their journey, possibly a farmer or someone who takes care of animals. The `<mask>` before this could be something like: "They meet a local farmer who shares his challenges."

→ Predicted: A complete other work.

**GPT-4o Output Second Run**
The text that should replace `<mask>` is: "Their first stop is Morocco."

→ Predicted: Yes, my mother is from Casablanca, and my father is from the town of Tinghir.

Figure 5: GPT-4o Infilling Analysis

In Fig. 5, we observe GPT-4o can only produce generalized insertions, which are not ideal as a query for retrieving a quotable segment. In contrast, our proposed model is trained specifically to learn to generate insertions from the context that includes detailed information that can be used as effective queries. Moreover, as the quotable segments (oftentimes an interview in our case) might be spoken by a different person other than the narrator, our proposed method can better capture the nuanced style of these quotable segments. We show an example on our demo page. GPT-4o's output is sometimes unreliable, making postprocessing challenging. While we believe GPT-4o could be improved with more contextual guidance in future work, our second-stage retriever remains effective.

# J    Quote Distribution

Table 9: Effect of the position of the `<SUM>` token

| Model | QCR (%) | QDI(mean ± std) (%) | Mean Absolute Difference (%) |
|---|---|---|---|
| Random Extraction | 98 | 11.71 ± 4.09 | 9.27 ± 3.66 |
| ETS[1] | 96 | 1.96 ± 0.49 | 1.71 ± 1.28 |
| A2Summ[4] | 96 | 3.98 ± 2.41 | 2.02 ± 1.77 |
| TeaserGen[11] | - | - | - |
| GPT-4o-DQ | 98 | 4.02 ± 4.09 | 3.49 ± 2.98 |
| GPT-4o-SP-DQ | 100 | 22.33 ± 8.18 | 19.88 ± 7.46 |
| REGen-DQ | 76 | 2.31 ± 2.12 | 2.18 ± 1.85 |
| REGen-IDQ | 67 | 1.98 ± 2.19 | 2.02 ± 1.93 |
| Oracle[2] | 76 | 2.45 ± 2.12 | - |
| Ground Truth | 82 | 3.02 ± 2.54 | - |

[1] ETS(extractive-then-smoothing) first extracts two interview segments, then uses GPT-4o to smooth them into a coherent output. To ensure fair comparison in downstream teaser generation, ETS is given prior knowledge that teasers typically contain two interview segments (based on the oracle average). Its closeness to the ground truth stems from this privileged information, effectively biasing it toward the oracle.
[2] Oracle: In our setting, the outputs generated by our dataset preprocessing pipeline, which includes pretrained models, are treated as oracle annotations representing an upper bound.

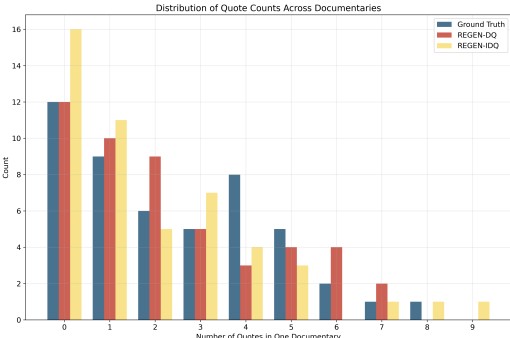

(a) The number of quotes per documentary, where the x-axis represents the number of quotes and the y-axis shows the count of documentaries

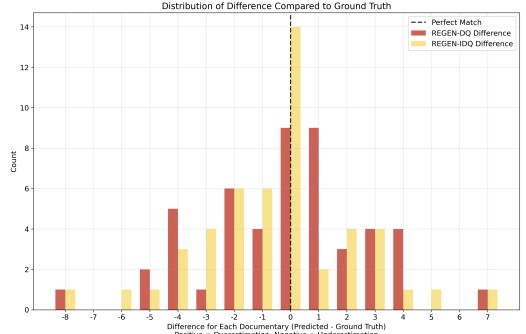

(b) Mean difference(can be negative) in quote count per documentary between the REGen system and the ground truth, with the x-axis as the mean difference and the y-axis as the number of documentaries

Figure 6: Detailed Distribution of Quotations in Our Generated Scripts

We present a more detailed distribution of quotations in our generated scripts in Table 9 andFig. 6 . From the table, we can see that REGen-IDQ is the closest to the ground truth distribution and is comparable to A2Summ[1]. In addition, we would like to point out that the position of the quotation marks and whether the retrieved content fits the context are also important in addition to matching the overall distribution.

# K    Survey Questions

## K.1    Demographic Questions

- How often do you watch documentary?
- Do you have any video editing experience?
- On a scale of 1–5, where 1 = Beginner and 5 = Native/Bilingual, how would you rate your English proficiency?

## K.2 Interview Insertion Evaluation

Please indicate your level of agreement with the following statement: *"The inserted interview integrates with the narration, effectively supports the surrounding claim, and maintains a natural flow."*

## K.3 Documentary Teaser Generation

- **Coherence**: To what extent do you feel that the sample maintains coherence and a smooth flow, ensuring that each segment transitions logically and the overall experience feels seamless?
- **Alignment**: To what extent do you feel that the narration and video match and work well together, making the overall presentation clear and easy to follow?
- **Realness**: How well do you feel this sample meets your expectations as a teaser for a documentary in general?
- **Interview Effectiveness**: Please indicate your level of agreement with the following statement: *"The inserted interview integrates with the narration, effectively supports the surrounding claim, and maintains a natural flow."*

## K.4 Generalizability Evaluation Prompts

- **Lecture Videos**: Which sample would you prefer as a teaser for lecture videos?
- **News Videos**: Which sample would you prefer as a teaser for news videos?

## K.5 Generalizability Evaluation Results

To assess the generalizability of our framework, we randomly select 10 lecture videos from the Multimodal Lecture Presentations Dataset [39], covering subjects such as psychology, machine learning, dentistry, and biology, and we also include full-episode news broadcasts from NBC News. We then conduct an A/B study comparing our method against TeaserGen [11], asking participants which teaser they would prefer for each lecture video or news broadcast. We perform a side-by-side evaluation against TeaserGen [11] on both lecture and news videos. In the lecture domain, participants prefer TeaserGen 62% of the time versus 38% for our model, indicating a clear preference for abstractive summaries there. For news videos, the split is 55% in favor of TeaserGen and 45 % for our approach—a smaller gap that does not reach significance.

Table 10: Subjective evaluation results of Generated Teaser

| Dataset | Model | Preference (%) |
| --- | --- | --- |
| News | REGen-IDQ-TV | 45 |
| News | TeaserGen | 55 |
| Lecture videos | REGen-IDQ-TV | 38 |
| Lecture videos | TeaserGen | 62 |

# L   Ablation Study

## L.1 Effects of Max Context Window Length

We compare different context window length in Table 11. We find no significant difference when we set different context window as the teasers are usually short.

Table 11: Effects of maximum context window length

| Model | Max context tokens | Recall@1 (%) | Recall@5 (%) | Recall@10 (%) |
|---|---|---|---|---|
| GPT-4o infilling | 128 | $2.78 \pm 0.48$ | $13.89 \pm 1.27$ | $22.50 \pm 1.44$ |
| GPT-4o infilling | 256 | $3.33 \pm 0.83$ | $13.33 \pm 0.83$ | $22.78 \pm 1.27$ |
| QuoteRetriever-T | 128 | 5.00 | 15.00 | 23.33 |
| QuoteRetriever-T | 256 | 4.17 | 16.67 | 22.50 |
| QuoteRetriever-TV | 128 | **5.00** | **17.50** | **30.00** |
| QuoteRetriever-TV | 256 | **5.00** | **17.50** | **27.50** |

## L.2 Effects of Alpha in Balancing Loss

We compare in Table 12 the effects of the weights of the generation loss and retrieval loss in the loss function Section 3.2. We find that when $\alpha = 1$, meaning that generation loss and retrieval loss are equally weighted, our model achieves its highest performance. In the following table, we set include 30% interviews in a batch as hard negative samples. Here we set max length of tokens for each sample being 256.

Table 12: Effects of $\alpha$ in the loss function Section 3.2

| $\alpha$ | Recall@1 (%) | Recall@5 (%) | Recall@10 (%) |
|---|---|---|---|
| 0 | 0.00 | 6.67 | 14.17 |
| 0.5 | 2.50 | 15.83 | 29.17 |
| 1 | 2.50 | 20.00 | 32.50 |
| 2 | 5.00 | 18.33 | 30.83 |

## L.3 Effects of Group Sampler

To examine whether treating interviews from the same documentary as hard negatives improves retriever training, we conduct two experiments. In the first, we treat all interviews within the same documentary as hard negative samples. In the second, we treat only 30 % of those interviews as hard negatives and omit the group sampler. We find that the 30 % group-sampler configuration yields the highest recall@5 and recall@10, while omitting the group sampler achieves the highest recall@1.

Table 13: Effect of negative sampler construction and loss function at $\alpha = 1$

| Model | Loss | GroupSampler | Recall@1 (%) | Recall@5 (%) | Recall@10 (%) |
|---|---|---|---|---|---|
| Model-Visual | Contrastive | ✓ | 5.00 | 17.50 | 27.50 |
| Model-Visual | L2 | ✓ | 1.67 | 6.67 | 13.33 |
| Model-Visual | Contrastive | 30% | 2.50 | 20.00 | 32.50 |
| Model-Visual | Contrastive | ✗ | 7.50 | 17.50 | 31.67 |

## L.4 Effect of Loss Function

We also use $L_2$ loss to find the closest embedding during the retrieval stage (see Table 13). We find that leveraging contrastive loss increases our model's performance, yielding higher recall. This is likely because contrastive loss can better differentiate embeddings that are close in the embedding space. Here we set max length of tokens for each sample being 256.

## L.5 Effect of Position of Retrieval token

As shown in Table 14, appending the special token <SUM> to the end of the decoder output before retrieval improves recall. Here we set max length of tokens for each sample being 256.

Table 14: Effect of the position of the <SUM> token

| Position | Recall@1 (%) | Recall@5 (%) | Recall@10 (%) |
|----------|--------------|--------------|---------------|
| Start    | 1.67         | 10.00        | 25.83         |
| End      | 2.50         | 20.00        | 32.50         |

## M   Comparison within REGen System

We compare the models in REGen system with different variants. We find that REGen-DQ achieves the highest ROUGE score and the most realistic quote distribution, as indicated by the quote coverage rate and quotation density index, both closest to the ground truth. The G-Eval scores for REGen-IDQ-T and REGen-IDQ-TV are closer to the ground truth than REGen-DQ, indicating that they produce more coherent, story-like scripts under automatic LLM evaluation. In Table 5, our subjective evaluation further indicates that our proposed models, REGen-IDQ-T and REGen-IDQ-TV, receive higher ratings for interview effectiveness compared with REGen-IDQ (random), indicating the effectiveness of our proposed retriever. In Table 4, we present the effects of different retriever methods in the documentary teaser-generation task, we find that REGen-IDQ-TV achieves the highest F1 score among models in REGen system. In Table 5 our subjective evaluation of teaser generation shows that teasers generated by REGen-DQ yield higher interview-effectiveness scores than those by REGen-IDQ-TV, indicating that fine-tuning to enable direct quotations can increase the coherence and supportiveness of inserted interviews.

