# OpenReview forum: "REGen: Multimodal Retrieval-Embedded Generation for Long-to-Short Video Editing"
_NeurIPS.cc/2025/Conference — NeurIPS 2025 poster_

### Official Review · Reviewer_sppt · 2025-06-18

**Clarity:** 3
**Significance:** 3
**Originality:** 3
**Rating:** 4
**Confidence:** 2

**Summary:**

This paper proposed REGen, a multimodal retrieval-embedded generation method for long-to-short video editing, which utilizes a large language model for multimodal resources quoting while maintaining a coherent narrative. The proposed method is evaluated on the task of documentary teaser generation and outperforms existing abstractive and extractive approaches in many evaluation metrics on teaser generation.

**Questions:**

See weaknesses. I believe that weakness 1 is especially crucial to address for theoretically supporting the evaluated performance. I would increase the rating if the reply is convincing.

**Ethical Concerns:**

["NO or VERY MINOR ethics concerns only"]

**Final Justification:**

The additional experiment solving weakness 1 is encouraged to have a deeper analysis in the final paper, such as adding a comparison on the newly provided Quote Distribution Plot between baselines and the proposed method.

Meanwhile, the main issues from the weaknesses of my concern have been solved. I will raise my initial rating to 4.

**Limitations:**

The authors adequately addressed the limitations and potential negative societal impact of their work.

**Paper Formatting Concerns:**

No major formatting concerns in this paper.

**Quality:**

2

**Strengths And Weaknesses:**

Strengths:
1. The overall idea is interesting, and the task is important for real applications.
2. The experiment result is solid, both objective and subjective evaluations are provided on quoting short interview clips within a coherent narrative.
3. Convincing visualization examples are provided to prove the effectiveness of the proposed method.

Weaknesses:
1. There is some controversy over the rationality of some evaluation metrics, for example, QCR and QDI may be close to ground truth due to the methods used that generate scripts with a similar quote distribution to the ground truth scripts. But it does not affirmatively mean that the closer QCR and QDI are to the ground truth, the better the method would be for generating a similar quote distribution to the ground truth. On the contrary, the paper should use the average absolute difference (of number of quotes inserted per documentary, or proportion of test videos in which at least one quotation is correctly inserted) between the method and the ground truth, to demonstrate the performance difference for generating scripts with a similar quote distribution to the ground truth scripts.
2. Recall of quote retrieval methods seems relatively low, but this should be considered as a core component in the framework of this article and deserves further analysis and improvement.
3. The complete framework of Figure 1 lacks some innovative method designs, which the author should incorporate with the method section of the article.

---

> ### Author Rebuttal · Authors · 2025-07-28
>
> We thank the reviewer for appreciating the significance of our proposed task, as also noted by reviewers yMCw and QQGE. We also appreciate your encouraging feedback on our extensive experiments, which reviewer tTNr similarly highlighted.
>
> **[Weakness 1]**:
>
> | Model             | QCR (%) | QDI (mean ± std)  | Mean Absolute Difference |
> |-------------------|---------|-------------------|--------------------------|
> | Random Extraction | 98      | 11.71 ± 4.09      | 9.27 ± 3.66              |
> | ETS \*           | 96      | 1.96 ± 0.49       | 1.71 ± 1.28              |
> | A2Summ            | 96      | 3.98 ± 2.41       | 2.02 ± 1.77              |
> | TeaserGen         | –       | –                 | –                        |
> | GPT-4o-DQ         | 98      | 4.02 ± 4.09       | 3.49 ± 2.98              |
> | GPT-4o-SP-DQ      | 100     | 22.33 ± 8.18      | 19.88 ± 7.46             |
> | REGen-DQ          | 76      | 2.31 ± 2.12       | 2.18 ± 1.85              |
> | REGen-IDQ         | 67      | 1.98 ± 2.19       | 2.02 ± 1.93              |
> | Oracle\**          | 76      | 2.45 ± 2.13       | –                        |
> | Ground Truth      | 82      | 3.02 ± 2.54       | –                        |
>
> Note:
>
> \* ETS (extractive-then-smoothing) first extracts two interview segments, then uses GPT-4o to smooth them into a coherent output. To ensure fair comparison in downstream teaser generation, ETS is given prior knowledge that teasers typically contain two interview segments (based on the oracle average). Its closeness to the ground truth stems from this privileged information, effectively biasing it toward the oracle.
>
> \** Orcale: In our setting, the outputs generated by our dataset preprocessing pipeline, which includes pretrained models, are treated as oracle annotations representing an upper bound.
>
> From the table, we can see that REGen-IDQ is the closest to the ground truth distribution and is comparable to A2Summ[1]. In addition, we would like to point out that the position of the quotation marks and whether the retrieved content fits the context are also important in addition to matching the overall distribution. We verify our performance through subjective evaluations (Please see table 3 and table 5) and present qualitative examples on our demo page (Please see section 3).
>
> In addition, we provide two comparison plots on our demo page (Section 0): (1) the number of quotes per documentary, where the x-axis represents the number of quotes and the y-axis shows the count of documentaries; (2) the mean difference(can be negative) in quote count per documentary between the REGen system and the ground truth, with the x-axis as the mean difference and the y-axis as the number of documentaries.
>
> **[Weakness 2]**:
>
> Thank you for pointing that out! Even though the recall is relatively low, our model is still able to retrieve quotable interviews that support surrounding claims while maintaining a cohesive narrative flow. This is further supported by our subjective evaluation (see Tables 3 and 5), where the effectiveness of interview insertion is significantly higher than the baseline. We also provide qualitative examples on our demo page (see Section 3). We hypothesize that this is because some interviews convey similar ideas—for example, multiple interviewees may be included to support the same claim. As a result, their representations in the latent embedding space are very similar, making it challenging to retrieve the exact original interview segment. However, this does not undermine the role of quotable interviews, which is to support the claims while preserving the overall narrative flow.
>
> **[Weakness 3]**:
>
> We will expand the quote retrieval and narration-matching sections in our camera-ready version. Specifically, we will elaborate on the quote retriever using the architecture shown in Figure 2. We will also expand the explanation of how narration-flow matching works. Please see our demo page for Figure 1.
>
> **Reference**:
>
> [1]He et al., Align and Attend: Multimodal Summarization with Dual Contrastive Losses, CVPR, 2023

---

> ### Comment · Reviewer_sppt · 2025-08-04
> **Respond to the author rebuttal**
>
> The additional experiment solving weakness 1 is encouraged to have a deeper analysis in the final paper, such as adding a comparison on the newly provided Quote Distribution Plot between baselines and the proposed method.
>
> Meanwhile, the main issues from the weaknesses of my concern have been solved. I will raise my initial rating to 4.

---

> ### Author Response · Authors · 2025-08-05
>
> Thank you for this valuable suggestion! We will include the Quote Distribution Plots in the final paper and provide a more in-depth analysis using both plots.

---

### Official Review · Reviewer_tTNr · 2025-06-28

**Clarity:** 3
**Significance:** 3
**Originality:** 3
**Rating:** 4
**Confidence:** 4

**Summary:**

This paper proposes a novel approach for generating short videos that combine a coherent narrative with embedded video excerpts from longer videos. While extractive summarization methods often fail to produce cohesive stories, existing abstractive methods cannot directly incorporate video clips from the source material. To address this, the authors introduce a retrieval-embedded generation framework (REGen). REGen first uses a fine-tuned large language model to generate a narrative script containing placeholders for video quotes, and then employs a retrieval model to select suitable video clips that replace these placeholders, ensuring alignment with the narrative. The method is evaluated on the task of documentary teaser generation, where short interview clips are commonly used to enhance storytelling. Results show that REGen effectively integrates video excerpts while maintaining narrative coherence, and outperforms existing extractive and abstractive approaches in terms of coherence, alignment, and realism, as demonstrated by both objective metrics and subjective user studies.

**Questions:**

1 Is the computation required by the proposed method expensive?

2 Have you ever had some ideas to improve the limitations you list?

**Ethical Concerns:**

["NO or VERY MINOR ethics concerns only"]

**Limitations:**

yes

**Quality:**

3

**Strengths And Weaknesses:**

Stengths:

**1**  The motivation of the paper is clear, that existing abstractive methods cannot directly incorporate video clips from the source material.

**2** The authors conducted extensive experiments, including thorough ablation studies and comparisons with baselines, which convincingly demonstrate the effectiveness of the proposed REGen framework.

Weakness:

**1 ** Although video insertions help improve factual grounding, the method can still mistakenly place quotes in incorrect contexts. Incorporating all quotable materials into the initial script generation could reduce this issue, but this is technically challenging for LLaMA-based models due to their limited context window.

**2 ** The approach relies on accurate video segmentation, such as speaker diarization, which may not be applicable in other domains like lecture recordings.

---

> ### Author Rebuttal · Authors · 2025-07-28
>
> We thank the reviewer and sincerely appreciate your acknowledgement of the strong motivation and significance of our work—also highlighted by reviewer yMCw—as well as the thorough experiments demonstrating our framework’s effectiveness. We address your comments point by point below.
>
> **[Question 1]**:
>
> **We run speaker diarization on a single A6000 GPU, and fine‑tune LLaMA and train the retriever (batch size 64) on a single A100 GPU.  The entire system was trained in under one day**. To build the candidate pool, we first separate each video’s audio into music, dialogue, and sound effects—providing a clean dialogue track that can enhance WhisperX’s accuracy. WhisperX then processes this track to identify speaker turns and determine start‑end timestamps for each quotable segment (the most time‑consuming step, yet still 50× faster than real time). Next, we extract textual and visual features for every interview segment using our feature extractor.
>
> Once the candidate pool is assembled, there’s no need to rerun diarization or feature extraction. Retrieval over L quotes and N candidates incurs O(L · N) time, so computation scales linearly with archive size and can become challenging for very large video collections. In our human‑annotated test set, each video contains on average three quotes—equating to roughly one second of retrieval time per video when inserting three video segments.
>
> **[Question 2, Weakness 1 and Weakness 2]**:
>
> As discussed in section 6, we identify two primary constraints. First, the limited context window can be addressed **by adopting a more powerful LLM with a larger context length**; we chose LLaMA for its strong open‑source performance, and our architecture readily accommodates such upgrades. Second, in lecture settings where the instructor’s voice is present (as discussed in the section 6), **we can use OCR‑based frame comparison to detect slide transitions and segment long lectures into smaller chunks**. This variation affects only the preprocessing stage, and our two‑stage method remains effective once quotable segments are extracted. **We hope this work will serve as a foundation for hybrid abstractive–extractive summarization that supports both factual grounding and narrative coherence, with significant potential in educational domains.**

---

> > ### Comment · Reviewer_tTNr · 2025-08-08
> > **Reply to authors**
> >
> > Thanks for your reply, which has addressed most of my concerns. But i decide to maintain my initial score1

---

### Official Review · Reviewer_QQGE · 2025-06-29

**Clarity:** 3
**Significance:** 2
**Originality:** 3
**Rating:** 4
**Confidence:** 3

**Summary:**

This work presents a new multimodal retrieval-embedded generation framework, named REGen, for editing long videos into shorts. It combines abstractive and extractive methods via a two-stage pipeline: 1) generating a story script with quote placeholders using fine-tuned LLaMA, and 2) retrieving supporting video clips from the input to fill these placeholders. Experiments on documentary teaser generation show REGen effectively inserts video quotes while maintaining narrative coherence, outperforming baselines.

**Questions:**

Please refer to the weaknesses for detailed concerns.

**Additional Questions:**
1. For videos without clear speaker segmentation (e.g., lectures), could REGen be adapted to detect "quotable moments" based on visual/audio cues (e.g., slide transitions, emphasis in speech) rather than relying solely on diarization?
2. How does the system ensure temporal coherence when inserting retrieved clips? For instance, if a selected interview clip references an event that has not yet been introduced in the narrative, could this lead to logical inconsistencies in the final output?

**Ethical Concerns:**

["NO or VERY MINOR ethics concerns only"]

**Final Justification:**

The authors' response has successfully addressed my concerns. I maintain my initial rating with higher confidence.

**Limitations:**

Yes

**Paper Formatting Concerns:**

No or very minor formatting issues

**Quality:**

2

**Strengths And Weaknesses:**

**Strengths:**
1. Effective framework for creating documentary teasers and has promising potential for applications in education and other domains.
2. The combination of extractive and abstractive summarization methods is inspiring.


**Weaknesses:**
1. The description of the methodology is overly dense and lacks emphasis on the novel innovations.
2. The choice of using fixed 10 chunks is not empirically justified or supported by ablation studies, leaving the rationale unclear.
3. Although the paper emphasizes maintaining narrative coherence while integrating factual quotes, the architecture relies on a sequential pipeline rather than joint optimization. This limitation is reflected in Table 6, where the overlap ratio results fail to fully support the claim of narrative coherence.

---

> ### Author Rebuttal · Authors · 2025-07-28
>
> We thank the reviewer for their constructive feedback. We’re encouraged that the review find our documentary‑teaser framework effective and our hybrid extractive–abstractive approach inspiring. We answer the reviewer’s comments below.
>
> **[Weakness 1]**:
>
> We propose a new retrieval-embedded generation framework that enables an LLM to quote multimodal resources while maintaining a coherent narrative. Our framework supports long-to-short video editing by generating short videos that feature a coherent narrative with embedded video insertions extracted from a long input video.
>
> **[Weakness 2]**:
>
> Thank you for pointing that out. Following TeaserGen [1], we split the transcribed text into smaller segments to generalize more effectively to extremely long videos.
>
> **[Weakness 3]**:
>
> We would like to clarify that the overlap ratio in Table 6 is used to measure the overlap between the content quoted by the large language model and the ground truth interviews. It is defined as {#overlap words} / {#words in matched interviews}. This metric is not used to evaluate narrative coherence. When training our retriever, we use joint optimization with a multitask loss:
>  L = $L_{gen}$ + α · $L_{ret}$. Here,  $L_{gen}$ encourages the model to attend to the context window around the masked segment, while  $L_{ret}$  helps differentiate relevant quotable segments from other candidates.
>
> **[Question 1]**:
>
> Speaker diarization is effective for content such as news and documentaries, where multiple speakers are involved. However, in lecture videos, a key constraint is that there is usually only one speaker—the instructor—making speaker diarization less applicable. To address it, we detect slide transitions using OCR to compare frames and identify the start and end timestamps of each slide. This allows us to segment long lectures into smaller chunks. Importantly, these adjustments only affect the data preprocessing step. Once quotable video segments are extracted, our two-stage method remains effective.
>
> **[Question 2]**:
>
> Yes, it could. We intend to alleviate this issue in our script generation stage and retrieval stage. In our script‑generation stage, we leverage an LLM to preserve temporal information. In our retrieval stage, the query embedding is generated from the surrounding context, thereby retaining temporal cues. In the future, we can incorporate these contextual embeddings into the representations of quotable interview segments in our candidate pool to ensure temporal coherence and maintain logical consistency.
>
> **Reference**:
>
> [1] Xu et al., TeaserGen: Generating Teasers for Long Documentaries, ICLR 2025.

---

> > ### Comment · Reviewer_QQGE · 2025-08-03
> > **RE: Rebuttal**
> >
> > Thank you to the authors for their response, which has successfully addressed my concerns. I will maintain my initial rating with higher confidence.

---

### Official Review · Reviewer_yMCw · 2025-07-02

**Clarity:** 4
**Significance:** 3
**Originality:** 3
**Rating:** 4
**Confidence:** 4

**Summary:**

This paper introduces REGen, a novel framework for generating short promotional videos (e.g., teasers) from long-form videos like documentaries. The core problem it addresses is the inability of existing methods to create a coherent abstractive narrative while also embedding ("quoting") verbatim clips from the original source to ground the story in facts.

REGen tackles this with a two-stage, hybrid approach with Script Generation followed by Quotation Retrieval.

The final output is a short video that combines a synthesized narrative with directly extracted clips. The authors demonstrate the effectiveness of this system on the DocumentaryNet dataset, showing through comprehensive objective and subjective evaluations that REGen outperforms purely extractive and abstractive baselines in generating teasers that are coherent, factually grounded, and realistic.

**Questions:**

I have a few questions on the empirical analysis below:

1. The *QuoteRetriever-TV* model fuses text and visual features. However, the objective results in Table 3 show that it does not significantly outperform the text-only *QuoteRetriever-T* on recall metrics. Could you elaborate on the role of the visual modality? Is it more important for qualitative aspects not captured by recall, or does its benefit depend on the specific type of content being retrieved?

2. The results show that prompting a powerful LLM like GPT-4o to fill in the quote content and then using that text for retrieval performs poorly. This is an interesting finding. Why do you think this approach fails? Does the LLM generate text that is too generic or semantically divergent from the actual interview clips, leading to poor retrieval?

3. The proposed pipeline involves multiple models, including calls to GPT-4o for some baselines and processing steps. Could you comment on the computational cost and scalability of the REGen system? Is it feasible for processing very large video archives, or is it better suited for single-document editing?

**Ethical Concerns:**

["NO or VERY MINOR ethics concerns only"]

**Final Justification:**

My concerns are addressed and I shall maintain the score.

**Limitations:**

I have covered potential empirical limitations in questions and weaknesses section above.

**Quality:**

3

**Strengths And Weaknesses:**

*Strengths*

1. The paper introduces a novel and highly relevant task: enabling generative models to "quote" multimodal content. This hybrid abstractive-extractive approach is a significant conceptual advance over existing video summarization techniques. It directly addresses a key weakness of purely abstractive methods (lack of factual grounding) and extractive methods (lack of narrative coherence), proposing an elegant and practical solution. The problem formulation is a strong contribution in itself

2. The REGen system is technically sound and well-engineered. The two-stage pipeline is a logical and effective way to decompose this complex task. The design of the Quote Retriever, which is jointly trained to perform masked infilling and retrieval, is particularly clever. The paper is supported by a thorough set of experiments with strong baselines.

3. The paper is exceptionally well-written and easy to follow. The motivation is clear, the methodology is explained in detail, and the figures (especially the system overview) are highly effective at illustrating the proposed framework.

*Weakness*

1. The system's performance is dependent on the quality of the initial speaker diarization used to identify quotable interview segments. The authors are transparent about the accuracy of this step, but errors here could negatively impact the entire pipeline by creating a flawed pool of candidate quotes. A brief analysis of the system's sensitivity to these upstream errors would be beneficial.

2. The method is developed and primarily evaluated on documentaries, a genre where the distinction between a narrator and an interviewee is often clear. The appendix includes a generalizability study on news and lecture videos, which honestly reports that users preferred a purely abstractive method for lectures. This suggests the "quoting" paradigm, as currently framed, may be best suited for specific content types.

3. The end-to-end system is a multi-stage pipeline involving several distinct models (ASR, diarization, LLM for scripts, retriever model). While this modularity is effective, it introduces complexity. Future work could explore more integrated, end-to-end models, although this would be a significant research challenge.

---

> ### Author Rebuttal · Authors · 2025-07-28
>
> We thank the reviewer for their thoughtful feedback and are encouraged by their recognition of our task’s novelty, the elegance and rigor of our hybrid abstractive‑extractive framework, and the clarity of our presentation. We hope this work meaningfully advances grounded and coherent long-to-short video editing. We answer the reviewer’s comments below.
>
> **[Weakness 1]**:
>
> Our system demonstrates robust speaker diarization performance on our documentary dataset, achieving a narrator‑prediction F1 score of 71.6%, start‑time correctness of 93.5%, end‑time correctness of 94.9%, and transcription accuracy of 94.9%. We thank the reviewer for pointing out the dependency of our framework on diarization quality. Although we acknowledge that alternative methods may further improve segment extraction on other datasets, **we evaluated our framework’s robustness** by injecting controlled noise: we introduced varying proportions of segments labeled as “narrator” into the set of quotable interview segments. **Using a fixed random seed, we then measured recall under different noise levels and reported in the table below; as expected, recall declines slightly as more incorrect segments are added.**
>
> | Model                              | Level of Accuracy                                                   | Recall @1 (%) | Recall @5 (%) | Recall @10 (%) |
> |----------------------------|--------------------------------------------------------|--------------:|--------------:|---------------:|
> | Ours                               | Build upon pretrained model                                         |          3.33 |         17.50 |          31.67 |
> | 10 % extra error   | Build upon pretrained model, intentionally add 10 % narration       |          1.67 |         14.17 |          29.17 |
> | 20 % extra error      | Build upon pretrained model, intentionally add 20 % narration       |          0.83 |         12.50 |          27.50 |
>
> **[Weakness 2]**:
>
> One possible reason is the preprocessing step used for lecture videos. As we mentioned in the limitations section, the current preprocessing method results in overly fragmented outputs in lecture videos. **We anticipate better performance by segmenting lecture videos based on slide transitions—for example, using OCR to detect transition points or leveraging built-in slide metadata that may accompany lecture recordings.**
>
> **[Weakness 3]**:
>
> Our pipeline begins by **inputing raw videos**: we first apply ASR and speaker diarization to convert audio into structured transcripts. Drawing on our video‑editing experience, we then use an LLM to infill and draft the script—essentially outlining where interview clips or other footage should be inserted—and follow up with a learned retriever to find the most appropriate video segments for each insertion placeholder. **This modular design not only mirrors real‑world editing workflows but also maximizes data efficiency by allowing each component to be trained with specialized and smaller datasets.**
>
> **[Question 1]**:
>
> In quotable interview segments, **the visual modality often provides background context**. For instance, an interview filmed outdoors differs from one conducted indoors. We observe that in current quotable segments—primarily interviews—the text modality carries more information than the visual modality. However, **our framework is designed to generalize to broader cases** where the visual modality contributes more substantially to retrieval. For example, when retrieving natural view videos without narration, visual information becomes important for accurate retrieval.
>
> **[Question 2]**:
>
> **We observe GPT-4o can only produce generalized insertions, which are not ideal as a query for retrieving a quotable segment**. In contrast, our proposed model is trained specifically to learn to generate insertions from the context that includes detailed information that can be used as effective queries. Moreover, as the quotable segments (oftentimes an interview in our case) might be spoken by a different person other than the narrator, our proposed method can better capture the nuanced style of these quotable segments. We show an example on our demo page. GPT-4o’s output is sometimes unreliable, making postprocessing challenging. While we believe GPT-4o could be improved with more contextual guidance in future work, our second-stage retriever remains effective.
>
> **[Question 3]**:
>
> **We run speaker diarization on a single A6000 GPU, and fine‑tune LLaMA and train the retriever (batch size being  64) on a single A100 GPU.  The entire system was trained in under one day**. To build the candidate pool, we first separate each video’s audio into music, dialogue, and sound effects—providing a clean dialogue track that can enhance WhisperX’s accuracy. WhisperX then processes this track to identify speaker turns and determine start‑end timestamps for each quotable segment (the most time‑consuming step, yet still 50× faster than real time). Next, we extract textual and visual features for every interview segment using our feature extractor.
>
> Once the candidate pool is assembled, there’s no need to rerun diarization or feature extraction. Retrieval over L quotes and N candidates incurs O(L · N) time, so computation scales linearly with archive size and can become challenging for very large video collections. In our human‑annotated test set, each video contains on average three quotes—equating to roughly one second of retrieval time per video when inserting three video segments.

---

### Note · Authors · 2025-08-13

We express our gratitude to the reviewers for their constructive feedback. In this paper, we propose a new retrieval-embedded generation framework that enables an LLM to quote multimodal resources while maintaining a coherent narrative. Below is a summary of the strengths highlighted by the reviewers and the concerns they raised, which we have addressed in the rebuttal.

**[Strengths Highlighted by Reviewers]**
1. **Novel and Significant Task**: Multiple reviewers noted the novelty and importance of enabling generative models to quote multimodal content, addressing key limitations of purely abstractive and extractive methods. Our method can serve as a foundation for hybrid abstractive–extractive summarization, supporting both factual grounding and narrative coherence.

2. **Effective and Inspiring Framework**: Reviewers highlighted the effectiveness of the REGen framework, including its clear two-stage design and the innovative Quote Retriever.

3. **Comprehensive and Solid Experiments**: Reviewers acknowledged the thorough evaluations, with strong baselines, objective metrics, and qualitative examples demonstrating the framework’s effectiveness.

**[Concerns Addressed in the Rebuttal]**

We have carefully considered the main concerns raised by the reviewers regarding the robustness, generalizability, and scalability of our proposed framework in the rebuttal. We will include all explanations and experiments added during the rebuttal in our final paper.

1. **Robustness and Generalizability**: We demonstrate the robustness of speaker diarization by evaluating the model's performance under varying levels of noise, showing that recall only slightly decreases as more incorrect segments are added. We also introduce alternative methods for obtaining quotable segments in different settings, such as using OCR to detect transition points or leveraging built-in slide metadata from lecture recordings. Importantly, these adjustments affect only the data preprocessing step—once quotable video segments are extracted, our two-stage method remains effective.

2. **Scalability**: We run speaker diarization on a single A6000 GPU and fine-tune LLaMA as well as train the retriever on a single A100 GPU with a batch size of 64. The entire model can be trained within a day. Furthermore, we show that the computational cost scales only linearly with the number of candidate quotable segments.

---

### Decision · Program_Chairs · 2025-09-17

**Decision:**

Accept (poster)

**Comment:**

All four reviews [yMCw,QQGE,tTNr,sppt] leaned towards acceptance and ultimately had the same borderline accept rating.

The reviewers appreciated several aspects of the work:

+ The motivation was considered clear [tTNr]
+ The idea was considered interesting [sppt]
+ The task was considered novel [yMCw] and important [yMCw,sppt]
+ The approach was considered a significant advance over existing video summarisation addressing a key weakness [yMCw]; similarly, the framework was considered effective [QQGE] and to have potential application in e.g. education [QQGE]
+ The combination of extractive and abstractive summarisation was appreciated [QQGE]
+ The proposed REGen system was considered technically sound and well engineered [yMCw]
+ The experiments and ablation studies were considered extensive [tTNr]
+ The experiment results were considered solid [sppt] and convincing [tTNr]
+ The visualisation examples were appreciated [sppt]
+ The paper was considered very well written and easy to follow by one reviewer [yMCw]

However, several weaknesses were pointed out, and authors responded to them in the rebuttal stage:

- The dependence of system performance on the quality of the initial speaker diarization was criticised [yMCw] and analysis of sensitivity to such upstream errors was desired [yMCw]; authors provided an experiment involving noise injection.
- Limitation primarily to documentaries was criticised [yMCw], as in appendix results on lectures a purely abstractive method was better; authors provided brief speculation that preprocessing of the lectures may be a cause of the difference.
- The complexity of the multi-stage pipeline was criticised [yMCw], and discussion of the cost and scalability was desired [yMCw,tTNr]; authors provided a brief argument about the pipeline and some details and discussion of the time consumption of the steps.
- Rationality of some evaluation metrics was criticised [sppt]; authors provided some discussion.
- Further analysis of low recall in quote retrieval was desired [sppt]; authors provided some discussion.
- Clarification on the role of the visual modality was desired [yMCw]; authors provided brief discussion.
- The claim of narrative coherence was considered not well supported by the sequential pipeline versus joint optimisation and the overlap ratio results [QQGE]; authors argued the overlap ration did not evaluate narrative coherence, and argued their multitask loss involves joint optimisation.
- Clarification of ensuring temporal coherence was desired with a concern about potential logical inconsistencies [QQGE]; authors replied they "intend to alleviate the issue" in the future via contextual embeddings.
- Analysis of the poor performance of LLM (GPT-4o) based quote filling-in was desired [yMCw]; authors analyse GPT-4o yielded generalised insertions.
- The use of fixed 10 chunks was criticised [QQGE]; authors replied they follow TeaserGen but did not seem to justify the amount of chunks.
- It was criticised the method could still place quotes in incorrect contexts [tTNr]
- Reliance on accurate video segmentation was criticised [tTNr] and its suitability to domains like lecture recordings was questions [tTNr]
- It was questioned whether the method could work by detecting moments instead of relying on diarization; authors provide some discussion on how in lecture videos they can detect slide transitions via OCR.
- Discussion on avoiding the limitations was desired [tTNr]; reviewers provided brief discussion of e.g. using OCR-based slide transition detection
- Discussion of some 'innovative method designs' were desired for the methods section [sppt]; authors stated they would expand some descriptions.
- One reviewer considered the methodological description dense and lacking emphasis on novelty [QQGE]; authors restated their main goal.

Ultimately after the rebuttals reviewers [yMCw,QQGE] considered their questions answered, [tTNr] considered most of the concerns addressed, and [sppt] considered their main issues to have been solved.

Overall, despite some concerns reviewers appreciated the work overall, and although some concerns could be further explored, it seems if the authors incorporate the material from their rebuttals the paper could be in a sufficiently good state to be presented at NeurIPS.